



# The DTU21 Global Mean Sea Surface and First Evaluation

Ole Baltazar Andersen[1], Stine Kildegaard Rose[1], Adili Abulaitijiang[2], Shengjun Zhang[3], Sara Fleury[4]

[1]DTU space, National Space Institute, Elektrovej 327/328, DK-2800 Kongens Lyngby, Denmark.
[2]University of Bonn, Institute of Geodesy and Geoinformation Nussallee 17, D-53115 Bonn, Germany
[3]School of Resources and Civil Engineering, Northeastern University, Shenyang, China
[4]LEGOS, Observatoire Midi-Pyrénées 14, avenue Édouard Belin 31400, Toulouse, France

*Correspondence to*: Ole B. Andersen oa@space.dtu.dk

**Abstract.** A new Mean Sea Surface (MSS) called DTU21MSS for referencing sea level anomalies from satellite altimetry is introduced in this paper and a suite of evaluations are performed. One of the reasons for updating the existing Mean Sea Surface is the fact, that during the last 6 years nearly three times as much data have been made available by the space agencies, resulting in more than 15 years of altimetry from Long Repeat Orbits or Geodetic Missions. This includes the two interleaved long repeat
cycles of Jason-2 with a systematic cross-track distance as low as 4 km.
A new processing chain with updated filtering and editing has been implemented for DTU21MSS. This way, the DTU21MSS has been computed from 2Hz altimetry in contrast to the former DTU15MSS/DTU18MSS which were computed from 1 Hz altimetry. The new DTU21MSS is computed over the same 20-year averaging time from 1993.01.01 to 2012.12.31 with a well-specified
central time of 2003.01.01 and is available from the following site;
(https://doi.org/10.11583/DTU.19383221.v1, Andersen, 2022)
Cryosat-2 employs SAR and SARin modes in a large part of the Arctic Ocean due to the presence of sea ice. For SAR and SARin mode data we applied the SAMOSA+ physical retracking in order to make it compatible with the physical retracker used for conventional Low-Resolution Mode data in other parts
of the ocean.

## 1 Introduction

Satellite altimetry provides highly accurate measurement of the ocean topography along the ground
tracks of the satellite (Fu and Cazenave, 2001; Stammer and Cazenave, 2017). For oceanography, the anomalous sea level about a mean reference surface is of primary interest. During the last two decades, Mean Sea Surface (MSS) as a reference surface has been developed with increasing accuracy (Pujol et al., 2018), Yuan et al, (2023)
To develop a MSS it would be optimal if observations were available on all time and spatial scales. The
challenge is to derive an MSS given limited sampling in both time and space using satellite observations. Another challenge is to merge repeated observations along coarse ground tracks with high spatial data from the geodetic mission (GM).



Thanks to new altimeter instruments and processing technology the accuracy of observed Sea Surface
Height (SSH) have increased dramatically over the last decade. It is important for deriving the Sea
Level Anomalies (SLA), that the reference or MSS is as accurate as the SSH in order to investigate
smaller mesoscale features (e.g., Dufau et al., 2016).

The paper is structured in the following way. Chapter 2 presents the details of the derivation of the new
DTU21MSS with focus on the improvement in data, retracking, processing and filtering. The chapter is
concluded with a subsection on the potential use of SAR altimetry from Sentinel-3A/B for the
DTU21MSS. Chapter 3 highlights various comparisons ranging from global comparison to regional
evaluations in the Arctic Ocean and for coastal regions illustrating the improvement in the DTU21MSS
model.

**2. Computation of the DTU21MSS**

The DTU21MSS is based on satellite altimetry data from frequently repeating Exact Repeat Missions
(ERM) and in-frequently missions with long or drifting repeat – called Geodetic Mission (GM). The
MSS is determined from a sophisticated combination of the coarse ERM with the high-density GM data
as described in Andersen and Knudsen (2008).
The first step is to select the averaging period and consequently the center time for the MSS. To enable
evaluations the agreement within the altimetric community has been to average over 1993.01.01 to
2012.12.31. Hence the center time for this and previous DTU models will be 2003.01.01. Within the 66º
parallels the highly accurate mean profiles derived using TOPEX/J1/J2 nearly uninterrupted
observations is the back-bone of the MSS models.
Table 1 shows all altimetry used for the computation of the DTU21MSS and its predecessors:
DTU15MSS and DTU18MSS. Whereas the DTU15MSS was based on roughly 5 years of GM
observations, the DTU21MSS is based on nearly three times as much data or more than 15 years of GM
due to the recent focus on prioritizing long repeat orbits.
It is also important, that satellite observations from the four newer GMs (Cryosat-2, Jason-1, Jason-2 &
SARAL) have around 1.5 times higher range precision compared with the old ERS-1 GM (Garcia et al.,
2014). Consequently it was decided to retire the older ERS1 and Geosat GM data for the DTU21MSS.

|  | Satellite | DTU15MSS | DTU18MSS | DTU21MSS |
|---|---|---|---|---|
| ERM | TP+Jason-1+Jason-2 | Jan 1993- Dec 2012 | Jan 1993- Dec 2012 | Jan 1993-Dec 2012 |
|  | ERS2+ENVISAT | May 1996-Oct 2011 | May 1996-Oct 2011 | May 1996-Oct 2011 |
|  | TP & Jason-1 Interleaved | Sep 2002 to Oct 2005 Feb 2009 to Mar 2012 | Sep 2002 to Oct 2005 Feb 2009 to Mar 2012 | Sep 2002 to Oct 2005 Feb 2009 to Mar 2012 |
|  | GFO | Jan 2001 Aug 2008 | Jan 2001 Aug 2008 | Jan 2001 Aug 2008 |
| GM | ERS1 (2 interleaved cycles of 168 days) | April 1994-May 1995 | April 1994-May 1995 | Not Used |
|  | Cryosat-2 (368.25 days repeat | Oct 2010-July 2014 | Oct 2010-July 2017 | Oct 2010- Oct 2019 |
|  | Jason1 LRO(1 cycle of 404 days) | April 2012-Jun 2013 | April 2012-Jun 2013 | April 2012-Jun 2013 |
|  | Jason2 LRO (2 cycles of 371 days) | Not used | Not used | Aug 2017-Sept 2019 |
|  | Saral AltiKa (drifting phase) | Not used | Not used | July 2016-Dec 2020 |

**Table 1:  Satellite altimetry used for the DTU15/18/21MSS models.**



The following sections describe the theoretical advances leading up to the release of the DTU21MSS compared with the previous DTU15MSS as well as other state of the art MSS models.
The first two advances related to short wavelength improvement where one advance is related to the retracking and filtering method used to enhance the short wavelength of the MSS and the second advance is related to the computation of new 2-Hz altimetric observations. The third and fourth
advances described are related to long wavelength corrections and the use of anisotropic filtering to enhance the MSS in current regions but also a new retracked Cryosat-2 dataset to enhance the Polar regions up to the 88 parallel.

**2.1 Satellite altimetry**

The Sensor Geophysical Data Record (SGDR) products for Jason-1 GM, Jason-2 GM, and SARAL/AltiKa GM are obtained from the Archiving, Validation, and Interpretation of Satellite Oceanographic (AVISO) data service. The L1b-level products for CryoSat-2 LRM are acquired through the data distribution service of the European Space Agency (ESA). All these products include along-track 20 Hz waveforms for all missions except for 40 Hz waveforms for SARAL/AltiKa.
All environmental and geophysical corrections of the altimeter range measurements have been applied to calculating SSH. These corrections include dry and wet tropospheric path delay, ionospheric correction, ocean tide, solid earth tide, pole tide, high-frequency wind effect, and inverted barometer correction. The most recent FES2014 ocean tide model has been used for all missions (Lyard et al., 2021). All corrections are provided on 1-Hz. Hence, these were interpolated into 20 Hz or 40 Hz by
using piecewise cubic spline interpolation.
All satellites except for CryoSat-2 operate in the traditional low-resolution mode (LRM) where the along-track resolution is limited to 2-3 km. Cryosat-2 also operates in LRM over most of the oceans. In regions where sea ice is prevailing Cryosat-2 operate in Synthetic Aperture Radar (SAR) mode. In this mode, the returning echoes are processed coherently resulting in a footprint of 290 meters. Over
steeply varying terrain and in some coastal regions, the SAR interferometric mode (SARin) is used where the instrument receives on two antennas are used. A mode mask controls the availability of three Cryosat-2 data types (www1, 2022). The advantage of the SAR processing is a near two-time range-precision improvement (Raney., 2011). Due to the burst structures of Cryosat-2, the improvement found is only around 1.5 times the range precision of LRM data. (Raney, 2011; Garcia et al., 2014)
Waveform retracking is an effective strategy to improve the range precision of altimeter echoes (Gommenginger et al., 2001). There are two strategies. Empirical retracker has the advantage of providing a valid and robust estimation of arrival time used to determine the SSH over almost all types of surfaces (e.g., sea ice leads, coastal). The disadvantage is, that empirical retrackers only provide SSH and not rise time used to determine significant wave height and windspeed. Hence its not possible to
determine the sea state bias correction to the SSH obsevations (Fu and Cazenave, 2001).

Physical retrackers generally apply the Brown model for LRM data (Brown, 1977) or the SAMOSA model for SAR and SAR-in observations (Ray et al., 2015). These estimate 3 or more parameters and



enable corrections and sea state conditions, through the determination of significant wave height and
wind speed. Hence these enable determination of sea state bias correction.

## 2.2 Two-pass retracking for range precision

Over the ocean, the waveforms from all four GM satellite missions are well-modeled and retracked
using the Brown-type model. In the first step, the waveforms are fitted by the three-parameter Brown
model (arrival time, rise time, and amplitude).
Maus et al., 1998 and Sandwell and Smith, 2005 demonstrated the presence of a strong coherence
between the estimation errors in the arrival time and rise time parameters resulting in a relatively noisy
estimate of arrival time and hence sea surface height. Consequently, Sandwell and Smith (2005)
suggested the use of a second step where the rise time parameter is smoothed. In the derivation of the
DTU21MSS, we applied the same two-step retracking and fixed the along-track smoothing at 40 km
before retracking the waveforms again using a two-parameter Brown model (arrival time and
amplitude).
For all four recent GM missions (Jason-1, Jason-2, SARAL/AltiKa, and CryoSat-2/LRM) this approach
has been proved effective (Garcia et al. 2014; Zhang and Sandwell 2017). Figure 1 illustrates the gain in
range precision using the two-pass retracking. The improvement for all four LRM datasets is dependent
on the SWH but is on average of the order of 1.5 similarly to what has been shown by other authors.
(Sandwell et al,, 2014; Zhang et al., 2019).

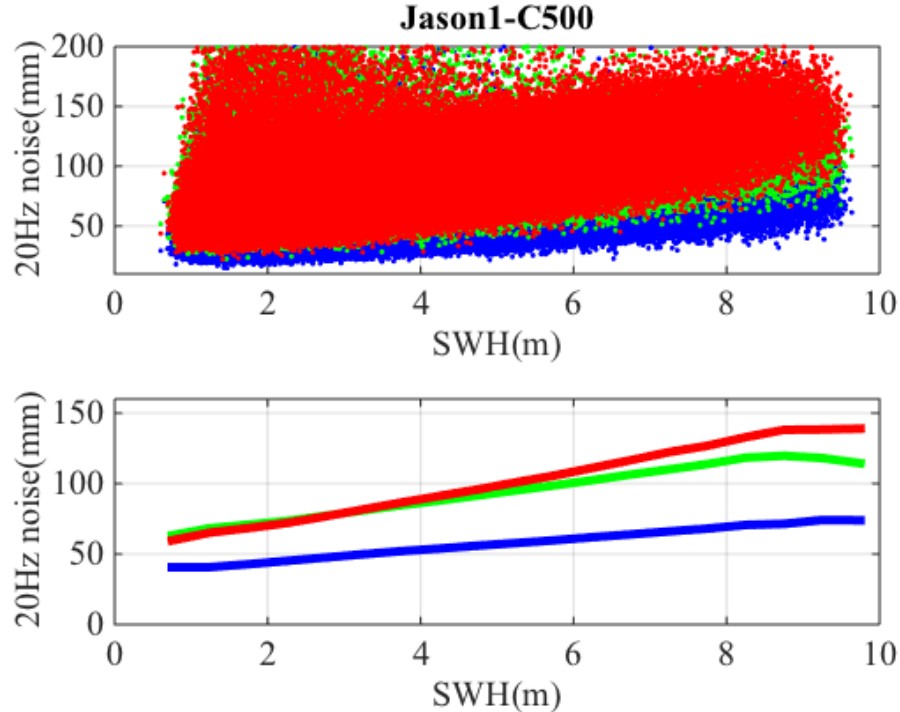

**Figure 1:  The standard deviation of retracked height with respect to DTU15MSS for cycle 500 (corresponds to the first 11 days of the Jason-1 GM). The upper figure illustrates the statistics for individual points. The lower figure illustrates the median averaged over 0.5 meters SWH intervals. Red: height from sensor geophysical data record; Green: height from the first step of two-pass retracking; Blue: height from the second step of the two-pass retracking). Modified from Andersen et al., (2021)**

Whereas two-pass retracking is very efficient for improving the range precision for the LRM data, we did not apply the two-pass retracking for the CryoSat-2 SAR- and SARin-mode data as there is no gain in range precision from the second step of the retracking for SAR and SARin data. This was documented by Garcia et al., (2014).

### 2.3. 2-Hz Sea Surface height data

The 20/40Hz double retracked SSH data are edited for outliers and subsequently, an along-track low-pass filtered is applied before generating the 2Hz SSH data used for the subsequent MSS determination.

The along-track low pass filter uses the Parks-McClellan algorithm which has a cut beginning at 10 km wavelength and zero gain at 5 km, thus the filter has 0.5 gain at 6.7 km, which is approximately the along-track resolution of 1-Hz data (Sandwell and Smith, 2009). The filter had to be designed for each satellite mission to match the 0.5 gain at 6.7 km due to the different along-track sampling rates. After this filter is applied the data were down-sampled to a 2-Hz sampling rate, which corresponds to an along-track spacing of around 3.3 km.

For the previous DTU15MSS we used 1-Hz SSH data from the Radar Altimetry Data Archive (RADS, Scharroo et al., 2013). In RADS, the 1-Hz data are computing as the average of all 20/40Hz data which is equivalent to use a boxcar filter. The advantage of using of the Parks-McClellan algorithm over the boxcar filter is, that this filter does not introduce side lobes degrading the SSH in the 10-40 km band contributing to the spectral hump of conventional LRM data (Dibarboure et al., 2014; Garcia et al., 2014). This is illustrated in Figure 2.

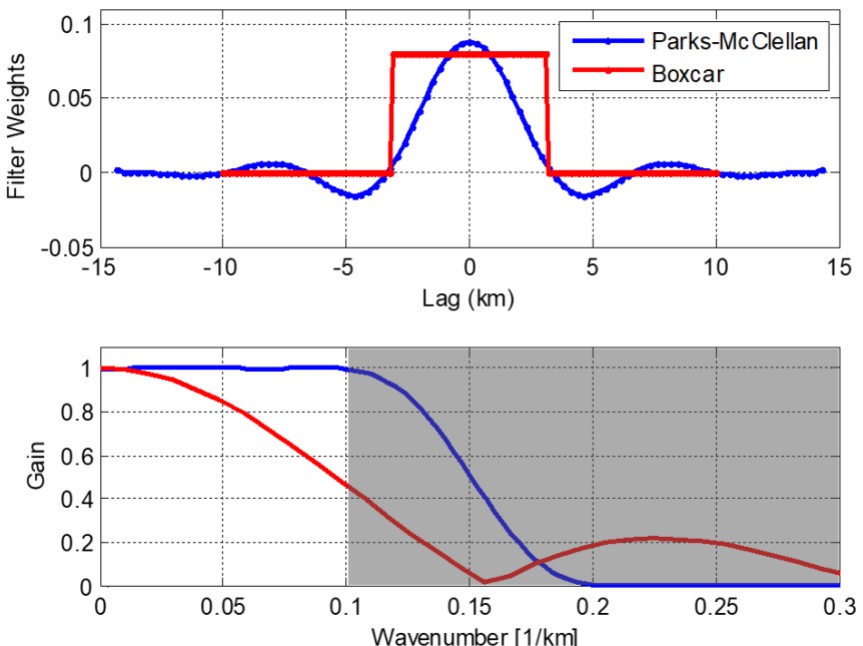

**Figure 2: Illustration of Parks-McClellan filter weights (blue) and the boxcar filter (red) to derive 1 or 2-Hz SSH data spatial filter (upper panelO. The lower panel illustrate the frequency response of the two filters. Sidelobes and spectral leakage in the 10-40 km wavelength can be seen for the boxcar filter, which will remain as high-frequency noise in the filtered dataset.**

### 2.4 Long-wavelength adjustment

The DTU21MSS builds on the heritage of the DTU15MSS. We first compute a long wavelength correction using the retracked and reprocessed ERM mean profiles. This is done separately inside the 66 ° parallel corresponding to mid and low latitude regions, where the TOPEX/J1/J2 are available and outside the 66 ° parallel where we have to rely on other satellites.



### 2.4.1 Mid and low latitudes

The long wavelength of the MSS within the 66° parallels, are largely defined by the highly accurate mean profiles derived using TOPEX/J1/J2 nearly uninterrupted observations every 9.91 days for 20 years. Along the mean profiles, the 2-Hz mean profiles are computed every 3 km, but across-tracks, the sampling is far less and up to 330 km at the Equator. Hence, significant spatial filtering has to be applied to the

The major ocean currents (e.g., the Gulf Stream and Kuroshio) flow largely west to the east giving rise to a significant Mean dynamic Topograpny signal which is also apparent in the MSS model. For DTU21MSS we introduced an-isotropic covariance function for the interpolation using least squares collocation (se Andersen and Knudsen, 2008). In the interpolation a second-order Gauss-Markov covariance model with a correlation length of 300 km in the longitude direction and 100 km in the latitude direction which was found to result in the best result. The small correction mainly focusing on the dynamic current systems is seen in Figure 3.

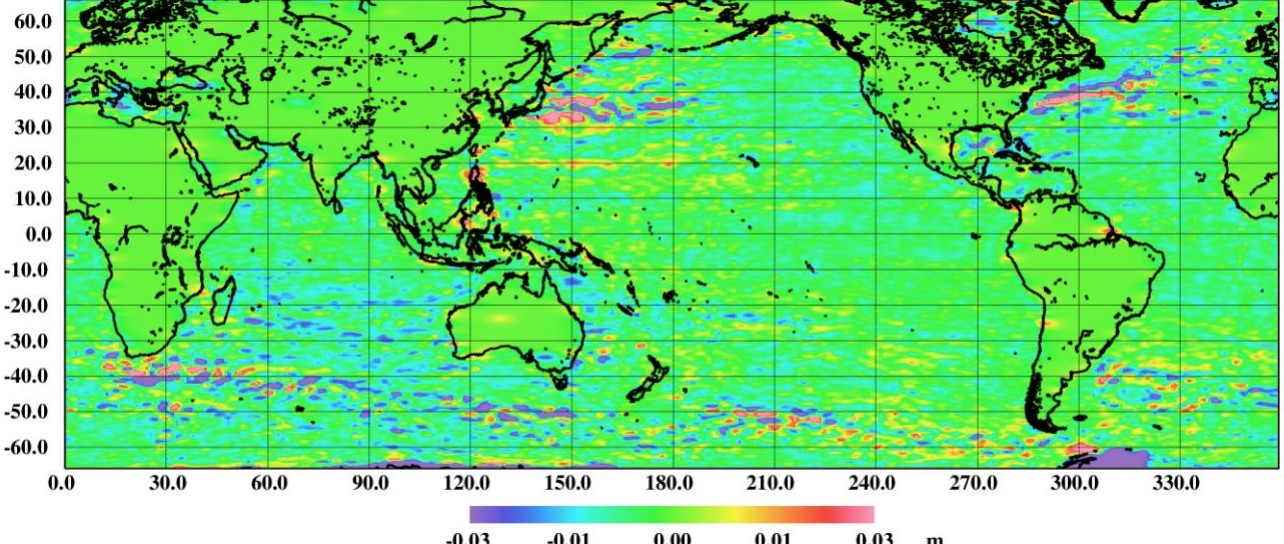

**Figure 3. The long wavelength correction to DTU15MSS computed from the TOPEX/J1/J2 mean profiles inside the 66° parallel.**

In the subsequent step the other mean profiles in Table 1 are introduced and adjusted to this model to derive the fine scales of the MSS model before the GM data are introduced. This follows the methodology described in detail in Andersen and Knudsen, 2008)

### 2.4.2. Polar region MSS from Cryosat-2

To improve the long-wavelength of the MSS outside the 66 parallels we used the Cryosat-2 which provides observations all the way to 88N. A closer inspection of the Cryosat-2 mode mask (www1,

2022) shows that Polar Regions (outside the 66 ° parallels) are largely measured in the SAR and SARin modes due to the presence of sea ice. This is with the exception of the Barents Sea north of Norway. For SAR and SARin mode data we applied the SAMOSA+ physical retracking (Dinardo et al., 2018). SAMOSA+ adapts the SAMOSA retracking model (Ray et al., 2015) to operate over specular scattering surfaces as ice-covered polar oceans by involving mean square slope as an additional parameter in the

retracking scheme and by implementing a more sophisticated choice of the fitting initialization resulting in greater robustness to strong off-nadir returns from land or else. The SAMOSA+ retracker even discriminates between return waveforms from diffusive and specular scattering surfaces, ensuring the continuity in the sea level retrieval going from the open ocean and into the leads in the sea-ice.

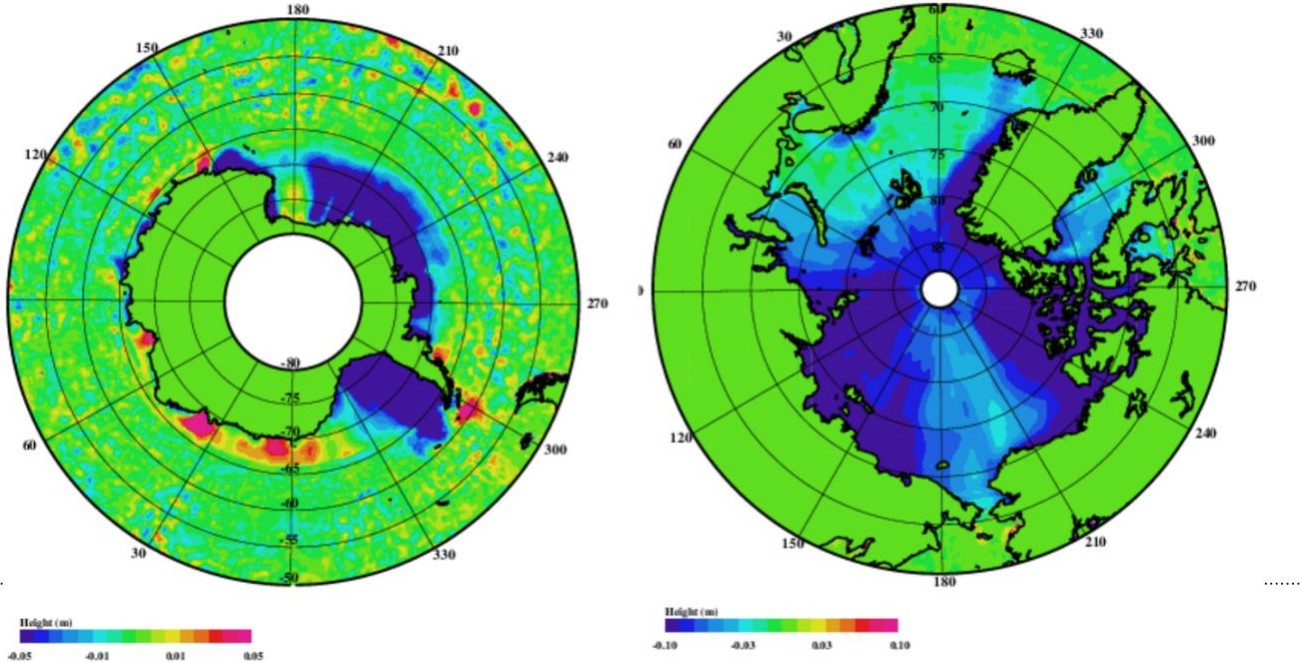


**Figure 4. DTU21MSS-DTU15MSS for the Southern Ocean (left) and the Arctic Ocean (right). The color scale ranges up to +/- 5 cm for the Southern Ocean and +/-10cm for the Arctic Ocean.**

With the assistance of the European Space Agency (ESA) Grid Processing On-Demand (GPOD) we

have processed a total of 9 years of Cryosat-2 (2010.10 to 2019.10) for both the Arctic and Southern Ocean using this SAMOSA+ retracker. Observations over the sea ice/open ocean interface were removed in the processing and only observations over leads (ocean surface between the ice floes) were selected similar to (Rose et al., 2019)

Upon computing mean profiles of Cryosat-2 observations, the center time for the Cryosat-2 data was

2015.04. It was found that it was necessary to correct for sea level rise to consolidate these data on the 2003.01 center period of the DTU15MSS and DTU21MSS following the methodology by (Rio and Andersen 2009). This was performed in the 65 ° - 66° border zone as the reprocessing of Cryosat-2 with SAMOSA+ is limited to outside the 65 ° parallels. This resulted in a correction of a few centimeters.



The difference between the DTU21MSS-DTU15MSS is shown in Figure 4 for both the Southern and
Polar Oceans. For nearly all ice-covered regions the DTU15MSS is higher than the DTU21MSS. We
expect this to be due to the fact that DTU15MSS was derived from 1-Hz RADS data which was very
sparse in both time and space. The few data in RADS is a consequence of tight editing and the fact that
RADS converts the SAR data to Pseudo LRM (Scharroo et al,, 2013) and performed physical retracking
on these data using a modified Brown model. In RADS we nearly only found data during the ice-free
summer month where the annual signal causes sea level to stand higher, so it is expected that
DTU15MSS could be biased high due to this.

**2.5 Mean sea surface computation**

The details of the computation technique of the DTU21MSS follows the development of former DTU
MSS models (Andersen and Knudsen, 2008) where the ERM tracks are first used to computed the
wavelength part of the MSS as shown in section 2.4. Hereafter the GM data are introduced to compute
the fine-scale structures of the MSS. This part uses small tiles to parallelize the computation process.
The final step to close the Polar Gap is to fill in MSS proxy data north of 88N where no altimetry is
available. This was done by feathering the EGM08 geoid (Pavlis et al., 2012) across the pole in the
following way: The preliminary MSS was calculated up to 88°N using the satellite altimetry data alone.
Subsequently, the difference between the MSS and the EGM08 geoid was computed longitude-wise in
the 87°N-88°N region and a mean offset was estimated and removed. The residual grid was transformed
into a regular grid in Polar stereographic projection enabling interpolation across the North Pole using a
second order Gauss Markov covariance function with a correlation length of 400 km. This makes the
DTU MSS models truly global.





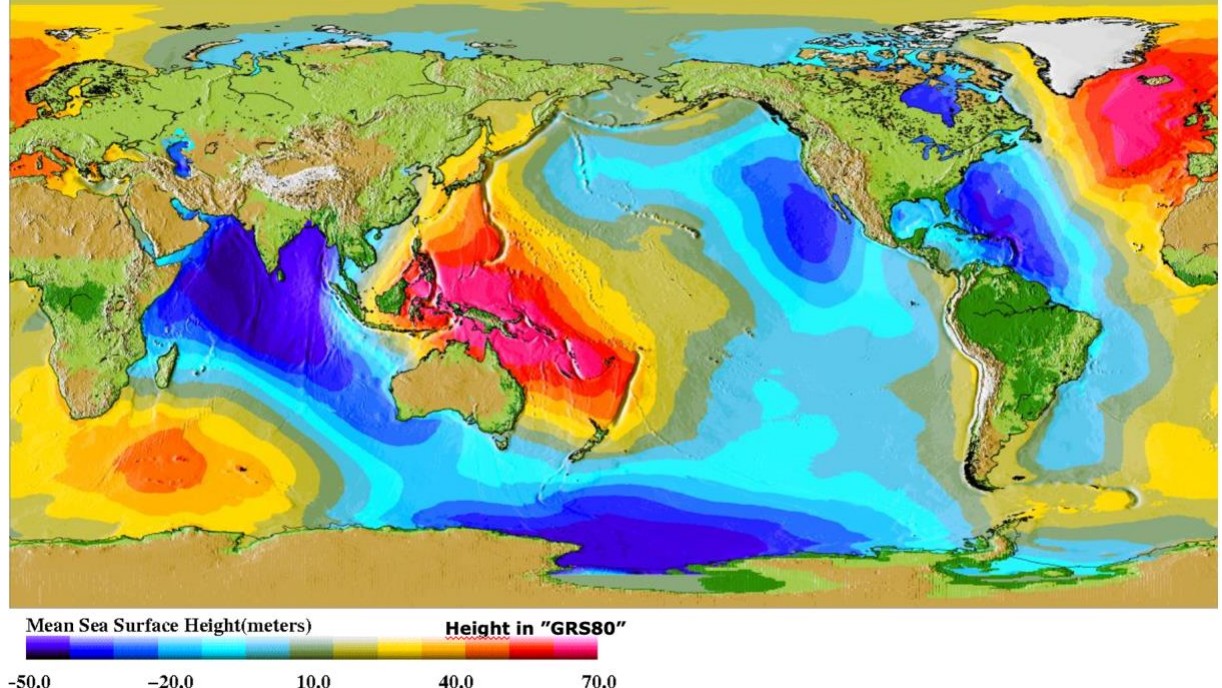

**Figure 5: The DTU21 mean sea surface from the Technical University of Denmark (DTU) in meters**


The DTU21MSS as its predecessors are all given on a 1-minute global resolution grid. A closer examination of the MSS in Figure 5 illustrates, that the height of the ocean's mean sea surface relative to the mathematical best fitting rotational symmetric reference system (GRS80) has magnitudes of up to 100 meters.


**2.6 Sentinel-3A/B SAR Altimetry**

The European Space Agency (ESA) launched Sentinel-3A on the 16th of February 2016 and Sentinel-3B on 25th April 2018. These satellites operate as SAR altimeters everywhere with the benefit of increased range precision compared with conventional LRM altimetry. Both the increased along-track resolution
and more importantly the improved cross-track resolution of 35 km for the combined Sentinel 3A/B dataset would make these important contributors to the DTU21MSS. However, two problems prevented the use of these data for the time being.
The first relates to the fact that mean profiles could only be computed over 5 and 3 years from Sentinel 3A and B, respectively. As the Sentinel-3 satellites operate in a 27-days repeat this resulted in as few as
66 and 40 cycles, making these mean profiles considerably noisier compared with other mean profiles. Secondly, the center times of Sentinel 3A/B is 2019 and 2020 which means that the mean profiles are more than 15 years away from the center time of the TOPEX/J1/J2 mean profiles. We try to illustrate the problem in Figure 6 showing a section of the Gulf Stream. The mean of S3A is 8 cm but the standard deviation of the spatial variation with respect to the DTU15MSS is as high as 13 cm (Figure 6
left panel). We show the mean profile from Sentinel-3A along track 719 (located at the blue arrow in



the left panel) across the Gulf Stream going from south to north (right panel of Figure 6). Between 26°N and 32°N the difference corresponds closely to the expected sea level rise of a little more than 8 cm. However, as the track crosses the Gulf Stream the signal increases to nearly 60 cm.

The mean dynamic topography associated with the Gulf Stream causes the mean sea level to drop by around a meter as one moves from the center of the Northwest Atlantic towards the coast. Due to the north/south meandering of the Gulf Stream it creates the observed sea level residual seen when the averaging period changes (Zlotniki, 1991).

As Sentinel 3A/B are both outside the (1993-2012) averaging period and as the meandering of the Gulf Stream is profound over the last 15 years, it was not possible to ingest the S3A and B mean profiles

without degrading the DTU21MSS in this region.

There is no doubt to the importance of Sentinel 3A/B for future MSS models, but in order to ingest the Sentinel 3A/B in future MSS models we found, that we will need to extend the averaging period to 30 years (1993-2022) to enable the use of these in future MSS models. We consequently decided to use the Sentinel 3A/B for the evaluation of the various MSS models.


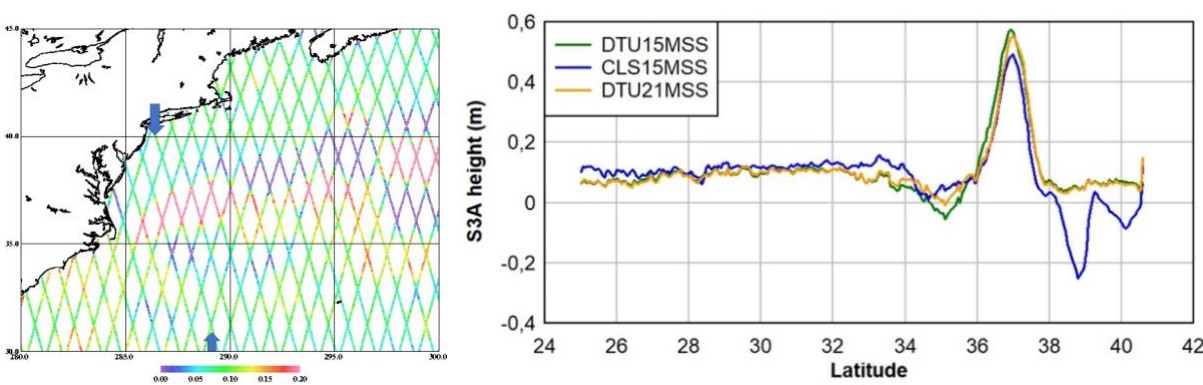

**Figure 6: Sentinel-3A 5y mean profiles in the Gulf Stream area (left) relative to the DTU15MSS. The Sentinel-3A mean profile for**
**track 471 (blue arrow) across the Gulf Stream relative to the DTU15MSS, the CLS15MSS (Schaeffer et al, 2012), and the DTU21MSS**

## 3. Evaluation

In this section, we perform three different evaluations of the MSS. These evaluations supplement the evaluation of previous MSS models performed by Pujol et al. (2018) and serve the purpose of indicating the improvements going from DTU15MSS to DTU21MSS globally, in the Arctic Ocean, and in coastal regions. The CLS15MSS is an improvement of the CLS11MSS (Schaeffer et al,, 2012) and is given on similar 1/60° resolution with similar averaging period to the DTU MSS models (Pujol et al., 2018).




### 3.1 Global evaluation with mean profiles

In the global evaluation we used data from the 1-Hz RADS data archive. These RADS data were used for the DTU15MSMS but not for the other MSS models. The global comparison in Table 2 illustrates the mean difference and the spatial variation when the mean profiles are spline interpolated onto the various MSS models. The zero offset and small standard deviation for the TP/J1/J2 mean profile is because all MSS are fitted to this profile in its derivation. The small offset for the other mean profiles corresponds to fact that the averaging of these profiles is not centered directly at 2003.01. The TP/J1/J2 and the TP/J1 interleaved are also used for the generation of all the MSS models. The increased spatial standard deviation correspond to the fact that far fewer repeat cycles are available for these mission (220 and 150 cycles, respectively) and the fact that these have been adjusted to the TP/J1/J2 in one way or the other.

|  | TP/J1/J2 (541936) | TP+J1 Interleaved (542638) | E2/ENV (1652043) | S3A (1446733) | S3B (1418477) |
|---|---|---|---|---|---|
| DTU15MSS | 0.00 /1.48 | 0.38 / 3.25 | -0.17 / 3.97 | 4.92 / 5.20 | 4.94 / 5.39 |
| DTU21MSS | 0.00 / 1.17 | 0.36 / 3.21 | -0.14 / 3.40 | 5.22 / 4.79 | 5.12 / 5.02 |
| CLS15MSS | 0.00 / 1.19 | 0.32 / 3.11 | -0.17 / 5.22 | 5.26 / 5.01 | 5.01 / 5.18 |

**Table 2: Comparison with mean profiles given as mean difference and standard deviation of spatial variations. All values are in cm.**

The Sentinel 3A and 3B mean profiles are independent of existing MSS models but only 66 and 40 cycle have been used, respectively. In the comparison with the Sentinal-3A/B mean profiles, we limited the comparison to within the 65° parallels. For all comparisons the number of repeat cycles can been seen to have a directly effect of decreasing spatial standard deviation with increasing number of repeat cycles. This illustrate the effect of natural variability of the sea surface and how this is gradually averaged out with increasing number of repeats. The 15 years or more different time-epoch between the S3A/B mean profiles and the center time of the MSS models directly illustrate the effect of global sea level rise during the altimetric era. All comparisons indicate that the DTU21MSS performs slightly superior compared with the older models.

### 3.2 Arctic evaluation.

Within the ESA CryoTempo project we evaluated the impact of the use of a physical retracker and empirical retracker on the retrieval of sea level anomalies in the Polar Ocean. We used the state-of-the-art empirical retracker called the Threshold First Maximum Retracker Algorithm (TFMRA) (Helm et al., 2014) and the SAMOSA+ physical retracker. In the evaluation, we also compared the state-of-the-art MSS models which were the DTU15MSS and DTU21MSS. It was not possible to include the CLS15MSS as this model only covers up to 84°N and has several voids in the Arctic Ocean (Pujol et al,, 2018). The use of the physical retracker allows us to estimate the Sea State Bias (SSB) which was



estimated. This Sea State Bias correction was subsequently applied to both the SAMOAS+ physical
SLA and the empirical TFMRA SLA.

A total of 7 months of Cryosat-2 was used between Oct 3013 and April 2014. The results are shown in
Figure 7 where the Upper panels show the spatial variation in the mean (two left panels for the TFMRA
and SAMOSA+ retracked SLA) and the corresponding standard deviation of SLA (two right panels).
The lower panels highlight the time evolution of the monthly SLA anomalies averaged with the monthly
mean given in the left panel and the standard deviation given in the right panel.

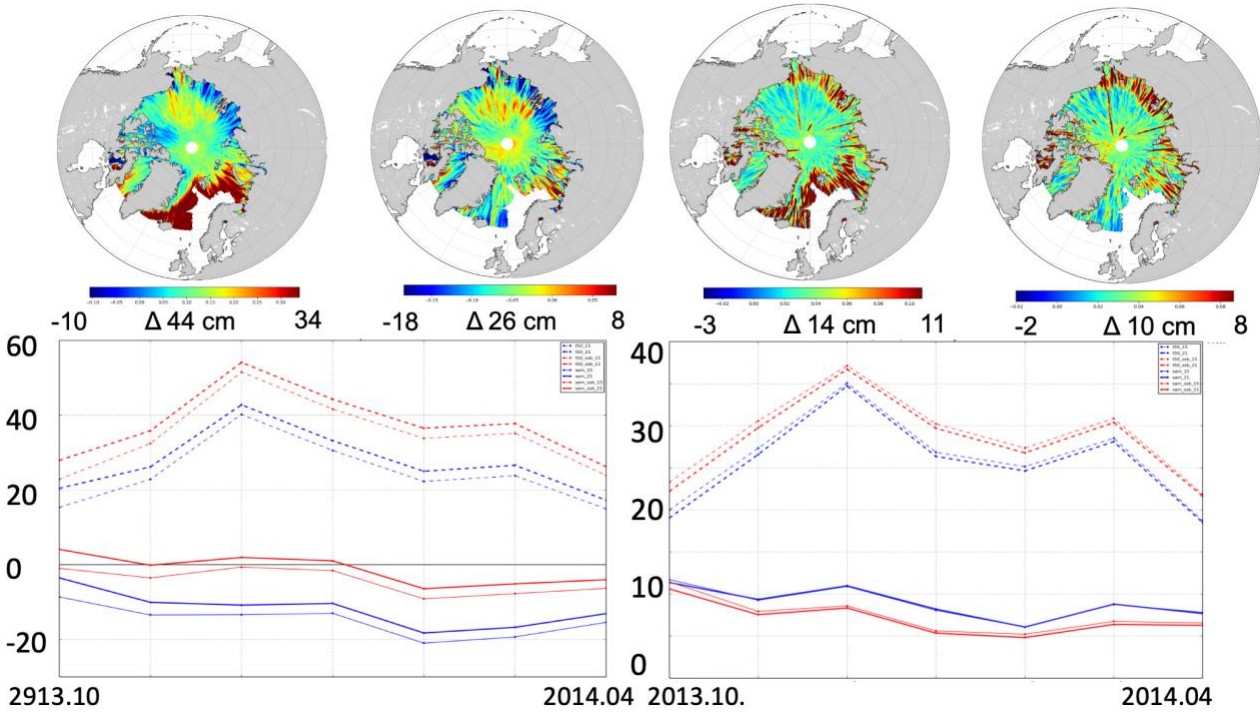

**Figure 7. Comparison of retrackers and MSS models over the Arctic Ocean from Oct 3013-April 2014. Upper panels: Mean SLA**
**using the empirical TMFRA retracker and DTU15MSS (first panel); Mean SLA using SAMOSA+ and DTU21MSS (second panel).**
**Standard deviation of SLA using the empirical TMFRA retracker and DTU15MSS (third panel) and standard deviation of SLA**
**using SAMOSA+ and DTU21MSS (fourth panel).**
**Lower panels: Evolution of SLA in time. Mean (left) and Standard deviation (right) shown as monthly values. Heavy lines**
**correspond to using DTU21 and thin lines correspond to using DTU15. Dotted lines correspond to using the TFMRA retracker**
**and solid lines to SAMOSA+ retracker. The red lines have the Sea State Bias correction applied whereas the blue lines have not.**

This study shows an improved measurement of SLA using the physical SAMOSA+ retracker and in all
cases, the DTU21MSS delivers better results than the DTU15 MSS. When using the physical
SAMOSA+ retracker er can see, that there is a clear effect of the ability to determine and correct for the
sea state bias (SSB). With SAMOSA+ sea state bias applied referenced to DTU21MSS we obtain a
mean SLA of -1.5cm ±12cm instead of -5.4cm±22cm over the 2013/10-2014/04 period when using an
empirical retracker and DTU15MSS




To illustrate the difference between various MSS models we computed the difference between the
DTU21MSS and the DTU15MSS and CLS15MSS, respectively.

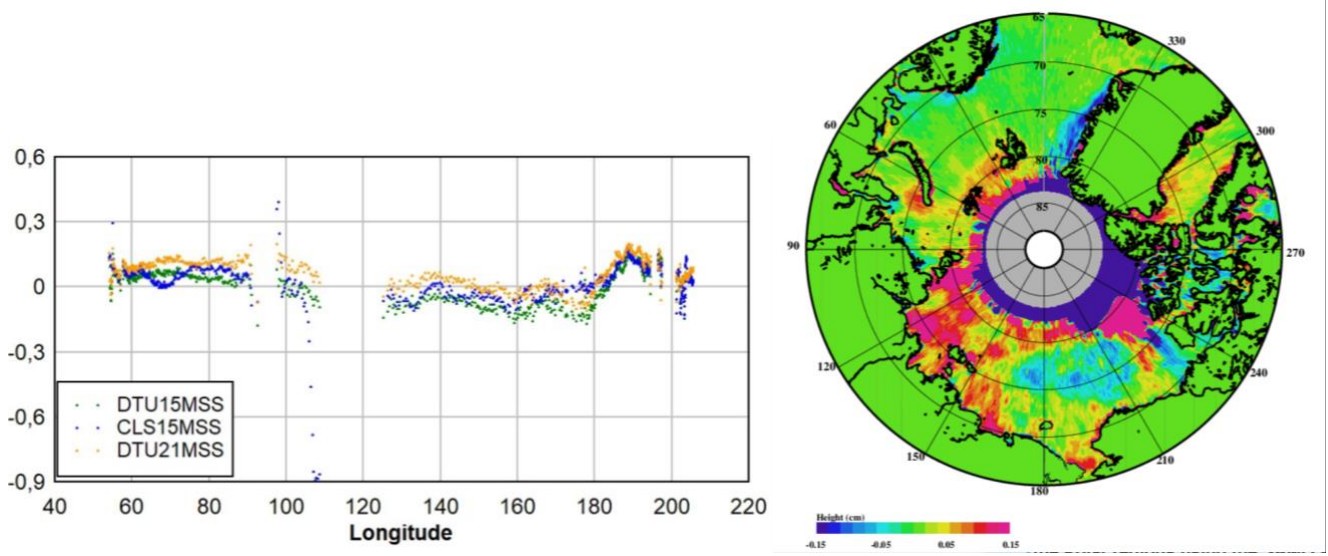

**Figure 8. The height difference (in meters) between the 5-year S3A mean profile along track 497/498 and various MSS models in the Arctic Ocean (left). Right: Mean Sea Surface difference between DTU21MSS and CLS15MSS Dark Blue Regions north of**
**Canada are voids in the CLS15MSS. The color scale ranges from -15cm to +15 cm.**

To illustrate the differences between the various MSS model we computed the difference with a
Sentinel-3A 5-year mean profile and the various MSS model. Figure 8 shows this difference along the
Sentinel-3A track 497/498. The track transits from Russia at 68ºN, 54ºE. Passing to the east of Nova
Zemlya and continues up to 82ºN (at 120ºE). From here it descends towards the Aleutian Trench at
57ºN, 204ºE. The standard deviation with the S3A mean profiles are 6.1 5.7 and 8.1 cm respectively for
the DTU15MSS, DTU21MSS, and the CLS15MSS. The missing data around latitude 90ºE is due to the
crossing of the Russian island Komsomoles. The missing data around 120ºE are due to voids in
CLS15MSS causing these data to be removed. The color scale ranges from -15cm to +15 cm. The
increase in the S3A residuals around 190°E is associated with the transition of the Bering Strait and the
in/out flow through the Strait (Woodgate and Peralta-Ferriz, 2021)

### 3.3 Coastal evaluation

The difference between the DTU21MSS and the DTU15MSS was evaluated in the Baltic Sea as part of
the BalticSeal+ project (http://balticseal.eu/). Differences are presented in Figure 9 and are ranging up
to 8 cm in the coastal zone and the narrow (15 km) Danish Straits as well as the Bay of Botnia and the
Swedish archipelago. In all locations we found, that the former DTU15MSS is unreasonably high near
the coastline. Similarly, we found that in the Bay of Finland the DTU15MSS was too low. In all cases,





we found that this is an artifact of the gridding combined with the lack of 1Hz data used for the older DTU15MSS.

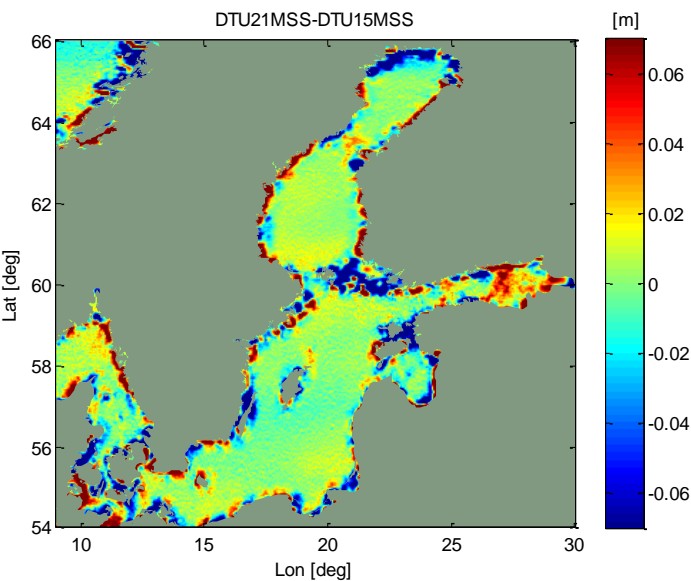

**Figure 9: The difference between the DTU21MSS and the DTU15MSS in the Baltic Sea including the opening to the North Sea**
**through the Danish Straits.**

## 4. Conclusions

A new Mean Sea Surface (MSS) called DTU21MSS for referencing sea level anomalies from satellite altimetry has been presented along with the first evaluations. We have presented the updated processing
chain with updated editing and data filtering. The updated processing filters the double retracked 20-Hz sea surface height data using the Parks-McClellan filter to derive 2-Hz sea surface anomaly. This Parks-McClellan filter has a clear advantage over the 1 Hz boxcar filter used for older DTU models in enhancing the MSS in the 10-40 km wavelength band. Similarly, the use of a the FES2014 ocean tide model improves the usage of sun-synchronous satellites in high latitudes in the new MSS.
Cryosat-2 employs SAR and SARin modes in large part of the Arctic Ocean due to the presence of sea ice. For SAR and SARin mode data we applied the SAMOSA+ physical retracking (Dinardo et al., 2018) in order to make it compatible with the physical retracker used for conventional Low-Resolution Mode data in other parts of the global ocean.

We initially performed global comparisons with the mean profile from various available satellite using
data from the RADS data archive as these have only been used in the DTU15MSS and not any of the other MSS models. The comparison with the independent  5- and 3-year S3A and S3B mean profiles show a relatively clear improvement for the DTU21MSS. This was also expected as the S3A/B satellites employs SAR altimetry and hence should compare better with the MSS derived using the two-pass altimetry due to the enhanced modeling of the 10-30 km wavelength (Garcia et al., 2013).



The evaluation in the Arctic Ocean clearly indicates an improved measurement of SLA using SAMOSA+ with the DTU21MSS. In conjunction with this physical retracker, the correction of the sea state bias (SSB) further improves the results. In all evaluations, the DTU21MSS delivers better results than the DTU15 MSS. With SAMOSA+, SSB, and DTU21MSS we obtain a mean SLA of -1.5cm ±12cm instead of -5.4cm±22cm over the 2013/10-2014/04 period.

Coastal evaluation of the new DTU21MSS was performed in the Baltic Sea and the Aleutian trench zone in Alaska. The evaluation in the Baltic Sea confirms that DTU15MSS is frequently several cm too high is coastal and Archipelago regions due to the lack of 1 Hz data for the DTU15MSS. The comparison with Sentinel 3A tracks close to the coast of the Aleutian. illustrated some oscillation problems with the CLS15MSS.

For the DTU21MSS we found that the 5-year Sentinel-3A mean profiles (2016.05-2020.05) were too problematic to consolidate onto the 1993-2012 averaging period without degrading the MSS model, particularly in large current regions. Consequently we omitted these data in the DTU21MSS, but also found that we shorth need to extend the averaging period to 30 years soon to enable the use of the important new Sentinel-3A/B data in the next-generation MSS models.


**Author Contributions**

OA wrote the manuscript and performed the computation of the DTU21MSS. ZS performed the two-pass retracking of all 20/40 Hz Geodetic Mission data. AA developing the software for producing 2 HZ and performed the MSS computations in coastal regions. SKR performed the data processing for SAR
and SARin data for the Polar Regions. SF contributed to the MSS validation in the Arctic Ocean.

**Funding**

ESA contributed to the MSS development through the Baltic+Seal project and the CryoTempo projects. SZ worked at DTU during 2020 supported by the National Nature Science Foundation of China, Grant No. 41804002, by the State Scholarship Fund of China Scholarship Council, Grant No. 201906085024,
by Fundamental Research Funds for the Central Universities.

**Acknowledgments**

The authors are thankful to the space agencies for considering the Geodetic or Long-repeat missions as part of mission operations and for providing these high-quality data to the users. We would like to acknowledge ESA-RSS (Research and Service Support), and in particular B. Abis and G. Sabatino, for
their assistance in processing the data with G-POD (http://gpod.eo.esa.int/). We acknowledge the support of ESA to the CryoTempo and Baltic Seal+ project through the contracts: AO/1-10244/20/I-NS & 4000126590/19/I-BG





**Data availability statement**

The DTU21MSS is available from http://data.dtu.dk. The high-resolution MSS model is available in several formats and relative to various reference ellipsoids (TOPEX and WGS84/GRS80) DOI: https://doi.org/10.11583/DTU.19383221.v1

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
