# Peer review of "The DTU21 Global Mean Sea Surface and First Evaluation"

_Earth System Science Data, 2023_

## Author Response (AR1)

Reviewer 1.

Thank for the very useful comments to the manuscript. I have replied to the following comments as they appear in the review. The review comments are shown in bold.

**Satellite altimeter data of 2 Hz instead of 1 Hz are used to construct DTU21MSS in the manuscript. The diameter of general footprint for radar altimeter is commonly about 4-10 km. So the correlation of 2-Hz SSHs should be considered in the study.**

Inded we have retracked the 20/40 Hz data in the derivation of the DTU21MSS using physical retracking. The radar altimeter footprint is larger for Ku band than Ka band. The trailing edge of the waveform which corresponds to distances larger than around 2.5 km from the center point of the radarpulse is mainly used to derive the SWH parameter. Hence there might be an effect this parameter. We do however use two-pass retracking which decouples the possible correlation.

**The time of DTU21MSS spans about 20 years (1993.1.1 to 2012.12.31). You know, 19-year is very better for the MSS time span (Yuan et al. 2020. https://doi.org/10.1016/j.csr.2019.104009). Many satellite altimeter data are collected since 2013, and it is very important to reduce them to 1993.1.1~2012.12.31 with the SSH time change correction method.**

There has been a lot of focus on the Accuracy of MSS models recently (e.g. Pujol et al., 2019) and particularly in the preparation for SWOT. I agree with the reviewer that we will soon need to update the MSS to encompass 30+ years of data. However we decided to keep the reference period of 20 years for the current MSS for the time being as this enables directly comparion between the DTU, CLS and SIO MSS models At line 66: we added: *We kept the averaging period for DTU21MSS to be able to validate the MSS directly with other MSS models. Changing the averaging period by as little as 3 years will change the mean by 1 cm as well as the spatial pattern due to ongoing sea level change (Veng and Andersen, 2019).*

**There are many satellite altimetry missions in table 1 with different SSH resolution and precision. How to evaluate effects of resolution and precision on MSS models? Here GM data are all since 2010, and how to reduce the time reference to 1993.1.1 to 2012.12.31?**

We updated the desciption how we deal with this at line 230.*The fine-scale computation is done in small tiles of 1° x3° with a 0.5 ° boundary to parallelize the computation process. As all wavelength longer that the size of the tiles are removed (roughly 200 km) we found that there was no need to adjust the period of the GM data to the MSS averaging period (1993-2012)*

**The uncertainty of MSS for the low frequency of over 150 km is very serious. And the uncertainty of MSS over the polar area and the coastal area is very great. How to evaluate and reduce these uncertainties?**

I think that we present the differences wiht other MSS models and indeed the difference is significant in Polar region. This is mainly due to the choice of retrackers as we also mention. This is also the reason why we investigate hugely computer ressources in the project to retrack all data in these regions. This has significantly reduced the errors when comparing with independent altimetry like S3A/B.

**What about the mechanism for the large change of MSS over the serious sea currents?**

In the computation of the mean profiles, we correct for the ocean variability as described in Pujol et a., 2017. I added the following sentence to explain at line 173: *In the computation of the mean profiles we apply the DUACS daily gridded sea level anomalies to remove ocean variability (Pujol et al., 2017)*

**Please increase practical applications of MSS model.**

We have added the following sentence to the introduction *Mean sea surface models are increasingly used as vertical offshore reference surfaces for offshore operations (e.g., dredging, windfarms, bathymetry surveys)*
* * *
Reviewer 2:

Thank you for the detailed review and constructive comments.

1. **There are multi-satellite altimetry data from 1993 to now. Why is the DTU21 MSS established by the altimetry data from 1993.01.01 to 2012.12.31?**

This is a very good and relevant question and we decided to update the manuscript to anwer and also make this clearer in the manuscript

Line 60 onwards; *There has been significant focus on the accuracy of MSS models (Pujol et al., 2019) in the preparation for the Surface Water and Ocean Topography (SWOT) mission. We consequently kept the same 20 years averaging period for DTU21MSS to be able to validate the MSS directly with other MSS models. Changing the averaging period by as little as 3 years will change the mean by 1 cm as well as the spatial pattern due to ongoing sea level change (Veng and Andersen, 2019).*

2. **The mean profile of TOPEX/J1/J2 is the reference for the DTU21 MSS. While the sea surface heights derived from TOPEX/J1/J2 are within the 66° parallels, those derived from other satellites, such as Cryosat-2 and Saral AltiKa, are outside the 66° parallels, so how do you unify the references between the two?**

This question was raised by both reviewers so we added a section at line 197 onwards to explain: *Upon computing the mean profiles of Cryosat-2 observations, the center time for the Cryosat-2 data was 2015.04. It was found that it was necessary to correct for sea level rise to consolidate these data on the 2003.01 center period of the DTU15MSS and DTU21MSS following the methodology by (Rio and Andersen 2009). This was performed in the 65 ° - 66° border zone as the reprocessing of Cryosat-2 with SAMOSA+ is limited to outside the 65 ° parallels. This resulted in a correction of a few centimeters.*

3. **Is the ERM data used for the DTU21 MSS construction also 2-HZ, and if so, how is it obtained, it is processed in the same way as the GM data?**

This is indeed a very good question. We added a section around line 66 to clarify this: *The long wavelength MSS was derived using the highly accurate nearly uninterrupted mean profiles derived using TOPEX/J1/J2. These data were taken from the 1 Hz data from the Radar Altimetry Data Archive (RADS, Scharroo et al., 2013). To extend the MSS into the polar regions outside the 66° parallel and to enhance the spectral resolution the other mean profiles shown in Table 1 from other Exact repeating satellites were fitted to the TOPEX/J1/J2 profiles. The differences were found by computing crossover differences between the ERM datasets. The crossover residuals were expanded into spherical harmonic degrees and order 2 to 4 and this surface was used to correct the ERM datasets. This methodology was similarly applied to derive DTU15MSS and DTU18MSS. Hence as a prior long wavelength model, we used a filtered version of the DTU18 MSS for wavelength greater than 100 km. For reference, the filtered version of DTU18MSS and DTU15MSS are virtually identical inside the 66° parallel.*

4. **Are both ERM and GM data processed by waveform retracted? If not, how is the ERM data corrected for the ocean sea state bias? If yes, where does TOPEX waveform data come from?**

This point is answered by the comment above as we used RADS for the long wavelength and hence we did not use the ERM waveforms for TOPEX (which we can not get anyway)

5. **What is the "correlation length", and why is it different in the longitude and latitude directions?**

We removed the section where this discussion appear and the associated Figure 3. This was removed, because that the entire discussion was tailored towards DTU18MSS which is not mentioned here. And when we computed differenced with DTU18MSS the signal in Figure 3 disappeared

6. **How can ERM and GM data be unified within the 66° parallels in the space-time reference?**

We added the following section at line 212 which explains that this is not such a bit issue as only short wavelength part of the GM data are used for the MSS computation:

*Line 212: The details of the computation technique of the DTU21MSS follow the development of former DTU MSS models (Andersen and Knudsen, 2008) where the ERM tracks are first used to compute the long wavelength part of the MSS as shown in section 2.2. Hereafter the GM data are introduced to compute the fine-scale structures of the MSS. The fine-scale computation is done in small tiles of 1º x3º with a 0.5 º boundary to parallelize the computation process. As all wavelengths longer than the size of the tiles are removed in this process (roughly 200 km) we found, that there was no need to adjust the period of the GM data to the MSS averaging period (1993-2012).*

7. **For Figures 4 and 6, why not compare DTU21 with DTU18?**

We decided to stick with the comparison with DTU15 in the figures. We could have added comparison with DTU18, but DTU15MSS and DTU18MSS would be identical in Figure 4.

In the Arctic presented in Figure 6 DTU18MSS was not preferred at it was determined using empirical retrackers as also discussed in the new section added at line 172: *For the polar regions we used the filtered version of DTU15MSS as a prior long wavelength reference. The reason is, that DTU18MSS was based on empirical retracked height in the*

*Polar regions. Frequently, physical and empirical retrackers differ in their height estimation in Polar regions (Rose et al., 2019). DTU15MSS was based on sparse physical retracked data from RADS. However, it was found to be a more consistent prior choice for DTU21MSS where physical retracking is used.*

 would have resulted in some leng And when we computed differenced with DTU18MSS the signal in Figure 3 disappeared

**8. Whether Figure 6 gives the mean profile from Sentinel-3A along track 719 or 471? Which is inconsistent with the statement on Line 260.**

Indeed this was a typo.

**9. The vertical coordinate to the right of Figure 6 should be SLA.**

This has been corrected:

**10. Figure 9 can only see the difference between DTU21 and DTU15 in the Baltic Sea and the Aleutian trench zone in Alaska, but cannot see that DTU15 is worse than DTU21.**

We removed the comparison in the Aleutian trench region but added a figure to illustrate the point raised by the reviewer. We demonstrated the better performance in the Baltic by comparing with the DVR90 which is fitted to 14 GNSS stations. Adding the following text:

*Line: 362 The difference between the DTU21MSS and the DTU15MSS was evaluated in the Baltic Sea as part of the BalticSeal+ project (http://balticseal.eu/). Differences are presented in Figure 8 (left) panels and are ranging up to 8 cm in the coastal zone and inside the narrow Danish Straits as well as the Bay of Botnia and the Swedish archipelago. In all locations we found, that the former DTU15MSS is unreasonably high near the coastline. Around the coast of Denmark, we further compared with the vertical reference frame model of Denmark called DVR90 (Web2, 2023). DVR90 is fitted to 14 GNSS stations along the coastline of Denmark. The right panel shows illustrate that DTU21MSS has a lower standard deviation close to the coast compared with DTU15MSS which independently verifies that DTU21MSS is superior in fitting Mean Sea Level close to the coast.*

**11. How did you get the long wavelength of the DTU21 MSS within the 66º parallels?**
**12. I don't understand the meaning of Figure 3.**

These two points are related. Principally the long wavelength of DTU21MSS is identical to the long wavelength of DTU18MSS inside the 66 parallel as this is used as prior model. Hence both the figure and discussion are obsolete and removed from the manuscript.

**13. From the overall DTU21 evaluation, it is not clear how much DTU21 has improved in the short wavelength band within the 66° parallels. It is also not clear how much the satellite altimeter data of 2 Hz instead of 1 Hz will improve the accuracy of the DTU21 MSS. Therefore, additional experiments should be needed to further evaluate DTU21.**

We decided not to change the manuscript to add more investigations besided the additional evaluation we performed in the Baltic Sea which also demonstrated the suporoity of DTU21MSS

This is partly because global evaluation of DTU21MSS has already been presented by an update by Pujol et al. presented at the OSTST, 2022).

We recently presented the the following computation of gravity anomalies derived from the DTU21MSS and compared this with the former gravity fields. Computation of the gravity anomalies directly reflect the improvement of short wavelength.

[Figure]

**Technical corrections have all been corrected according to suggestions.**